# Making genomic surveillance deliver: A lineage classification and nomenclature system to inform rabies elimination

**Kathryn Campbell**[1]*, **Robert J. Gifford**[2], **Joshua Singer**[2], **Verity Hill**[3], **Aine O'Toole**[3], **Andrew Rambaut**[3], **Katie Hampson**[1], **Kirstyn Brunker**[1]

**1** Institute of Biodiversity, Animal Health and Comparative Medicine, University of Glasgow, Glasgow, United Kingdom, **2** MRC-University of Glasgow Centre for Virus Research, University of Glasgow, Glasgow, United Kingdom, **3** Institute of Evolutionary Biology, University of Edinburgh, Edinburgh, United Kingdom

* k.campbell.1@research.gla.ac.uk

**Data Availability Statement:** All files are available from GitHub: https://github.com/KathrynCampbell/RABV_Lineages.

## Abstract

The availability of pathogen sequence data and use of genomic surveillance is rapidly increasing. Genomic tools and classification systems need updating to reflect this. Here, rabies virus is used as an example to showcase the potential value of updated genomic tools to enhance surveillance to better understand epidemiological dynamics and improve disease control. Previous studies have described the evolutionary history of rabies virus, however the resulting taxonomy lacks the definition necessary to identify incursions, lineage turnover and transmission routes at high resolution. Here we propose a lineage classification system based on the dynamic nomenclature used for SARS-CoV-2, defining a lineage by phylogenetic methods for tracking virus spread and comparing sequences across geographic areas. We demonstrate this system through application to the globally distributed Cosmopolitan clade of rabies virus, defining 96 total lineages within the clade, beyond the 22 previously reported. We further show how integration of this tool with a new rabies virus sequence data resource (RABV-GLUE) enables rapid application, for example, highlighting lineage dynamics relevant to control and elimination programmes, such as identifying importations and their sources, as well as areas of persistence and routes of virus movement, including transboundary incursions. This system and the tools developed should be useful for coordinating and targeting control programmes and monitoring progress as countries work towards eliminating dog-mediated rabies, as well as having potential for broader application to the surveillance of other viruses.

## Author summary

The importance of the ability to track the diversity and spread of viruses in a universal way that can be clearly communicated has been highlighted during the SARS-CoV-2 pandemic. This, accompanied with the increase in the availability and use of pathogen sequence data, means the development of new genomic tools and classification

**Funding:** This work was supported by The University of Glasgow (MVLS DTP studentship 125638-06 https://www.gla.ac.uk/colleges/mvls/graduateschool/mvlsdtp/) to KC and the Wellcome Trust (research grant 207569/Z/17/Z https://wellcome.org/) to KH. The funders had no role in study design, data collection and analysis, decision to publish, or preparation of the manuscript.

**Competing interests:** The authors have declared that no competing interests exist.

systems can strengthen outbreak response and disease control. Here, we present an easy-to-use objective and transferable classification tool for tracking viruses at high resolution. We use rabies virus, the cause of a neglected zoonotic disease that kills around 59,000 people each year, as an example use case of this tool. Applying our tool to a global clade of rabies virus, we find an increase in resolution from 22 subclades to 96 lineages; this fourfold increase in the definition at which we can classify the virus, may allow us to identify areas of persistence and transmission that were not previously apparent and further resolve patterns of virus spread. Insights from the application of this tool should prove valuable in targeting dog vaccination campaigns and improving surveillance as countries work towards the elimination of dog-mediated rabies.

## Introduction

Rabies virus (RABV) causes around 59,000 deaths and costs in excess of $8.6 billion per year [1], with a near 100% mortality rate after the onset of symptoms [2]. Post-exposure prophylaxis is highly effective in preventing rabies if administered quickly following exposure [3] but this is not always possible given its high cost and limited accessibility. Rabies can occur in all species of mammal, but up to 99% of human rabies cases arise from bites from infected domestic dogs [4,5]. Vaccinating dogs to interrupt transmission is therefore paramount [6] and a major focus of 'Zero by 30', the global strategy to eliminate human deaths from dog-mediated rabies by 2030 [7]. The focus of 'Zero by 30' is on dog-mediated rabies, but spillover from dogs into other carnivores often occurs, generally causing only short-lived chains of transmission [8]. Genomic surveillance is therefore vital to identify and monitor any wildlife reservoirs that may emerge [9].

To achieve the 'Zero by 30' goal, effective and coordinated surveillance is essential. Genomic surveillance can complement routine epidemiological surveillance through the insights it can provide on the lineages circulating in an area and any sources of incursions [10]. Sequence data proved useful in understanding rabies dynamics in Bangui, the capital of the Central African Republic [11]. Instead of sustained transmission in the city, local extinction occurred on three occasions with introductions from surrounding areas reseeding circulation, showing the need to expand control efforts across a larger geographic area [11]. Genomic surveillance can also reveal host shifts and other unusual dynamics. When rabies in Arctic foxes underwent a host shift to infect red foxes (*Vulpes vulpes*), the virus spread rapidly throughout Canada, resulting in epidemics in the 1950s and 1960s before being apparently eliminated in 1990 [12]. However, between 2015 and 2017 a number of rabies cases occurred in wildlife which were genetically similar to sequences from these historic fox rabies epidemics. Analysis revealed that although several lineages were eliminated, one persisted and underwent a host switch to skunks [12].

To monitor how an epidemiological situation is changing first requires characterization of the current situation. Several well-defined RABV clades circulate globally, within two major phylogenetic groups; bat-related and dog-related [13]. The dog-related group is split into between 5 and 7 different clades, depending upon the classification [13,14]. This includes the Cosmopolitan clade for which there are over 9500 publicly available sequences (including sequences of all lengths) split into 22 subclades, present in over 100 countries [13,15]. These discrepancies in clade numbers, however, are illustrative of issues surrounding the interpretation and classification of RABV phylogenetic data as a universal

naming system does not extend past high-level classification, with no set rules for defining lineages. Additionally, genomic data availability varies. Increasingly studies are focusing on whole genome sequences (WGS) given the greater resolution they provide, but the vast majority of studies thus far have generated shorter, partial gene sequences [15–17]. A lineage designation system therefore needs to be able to incorporate all of these data to provide maximal contextual information. Relevant terms needed to understand a lineage designation system are defined in Box 1.

## Box 1

### Definitions

**Clade**–Monophyletic group of sequences with a single ancestor; already defined by various studies for RABV [13,18].

**Subclade**–A smaller monophyletic group contained within a larger clade; again, defined in previous RABV studies [13].

**Cosmopolitan Clade**—A globally distributed, diverse clade of rabies virus that contains 22 subclades (Fig 1) [13].

**Lineage**–A group of genetically related sequences defined by statistical support of their placement in a phylogeny and genetic differences from a common ancestor. Clades and subclades (illustrated in Fig 1) can be split further into lineages which is typically a higher resolution classification. These are not yet defined for RABV—but Cosmopolitan clade lineages are defined in this study.

 **Major lineage**–A lineage named with a letter—e.g. A1 or Cosmopolitan AF1b_A1, that can be the first iteration of a lineage, or a lineage that has undergone significant evolution to become a new major lineage (Fig 1).

 **Minor lineage**–A lineage named with numbers (following the major clade nomenclature)—e.g. A1.1.1 or Cosmopolitan AF1b_A1.1.1 This is a major lineage that has undergone evolution, but not enough to become a new major lineage.

**Lineage designation**—An initial step to designate lineages based on a set of reference sequences, or the first set of data from an area, defined by an existing set of rules, and to name them accordingly. This is completed once to form a reference set of sequences to be used for lineage assignment, and may need to be updated as genetic diversity accumulates or is generated by sequencing efforts.

**Lineage assignment**—Identifying which existing lineage, defined by the initial lineage designation step, a new sequence belongs to.

**MAD DOG**—Method for Assignment, Definition and Designation Of Global rabies virus lineages; the method and corresponding tools presented in this study.

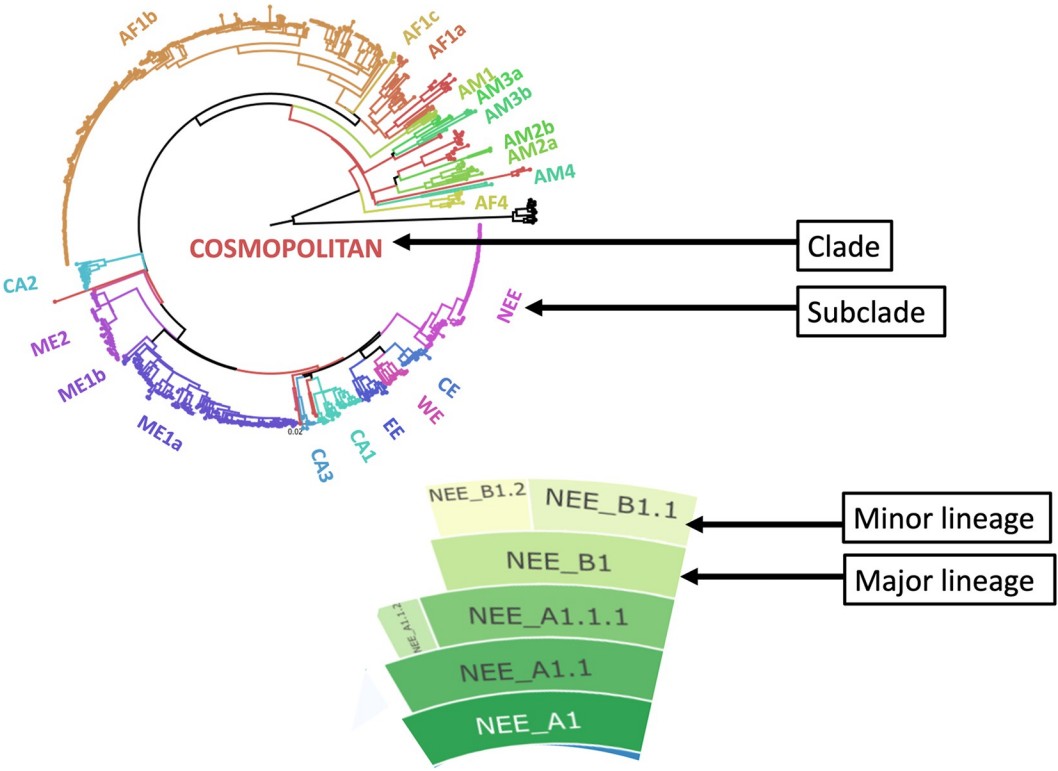

**Fig 1. Illustration of rabies virus clades, subclades and lineages.** Maximum likelihood tree of all publicly available whole genome rabies virus sequences (n = 650) coloured by previously identified subclades [13]. The NEE subclade has been expanded to show major and minor lineages from the updated MADDOG classification system.

Although the current phylogenetic classifications generally work well at a global level, they lack resolution for surveillance at more local or regional scales. The existing system generally represents where the different clades originally emerged and their early geographical distribution [13]. However, these clades do not always remain isolated to particular areas, and their dispersal is affected by human movements, leading to frequent introductions of lineages to new areas and their subsequent co-circulation [10]. Moreover, control and elimination efforts will further affect the distribution and diversity of circulating lineages. In many regions, a limited number of subclades appear to circulate, therefore, without a higher resolution classification it becomes difficult to identify patterns like lineage extinction that are to be expected, and therefore to monitor the impacts of control.

Rambaut et al. (2020) propose a universal virus nomenclature system to address these issues, which they apply to SARS-CoV-2 [18]. The system outlines a set of rules for classifying and naming viral lineages to produce a standardised tool to identify viral diversity on global and local scales, to track lineage emergence and transmission, and to allow for coherent updates as lineage turnover occurs [18]. Here, we adapt and apply these rules to a large diverse major clade of RABV to produce an updated, dynamic, classification system for the virus. The increased definition provided allows for more detailed genomic surveillance that can be used to monitor the circulation of viral lineages and identify incursions, unusual transmission routes and potential host shift events.

## Methods

### Data collection and processing

All available RABV sequences (n = 23,386), irrespective of sequence length and clade, and their associated metadata were downloaded from the RABV-GLUE (Box 2) resource via the website (http://rabv.glue.cvr.ac.uk). Many metadata entries stripped from Genbank and added to the RABV-GLUE database were missing information necessary for analysis, including the sample collection year, host, and location (n = 9650). Records with missing information were searched manually and any information that could be found from primary publications was updated into RABV-GLUE. The dataset of all sequences was then filtered to only include sequences designated to the Cosmopolitan clade by RABV-GLUE, excluding vaccine strains. For the purposes of these analyses we considered WGS, covering >90% of the genome (at least 10,000 nucleotides (nt)), and nucleoprotein (N) gene sequences (1300 nt), thus excluding smaller partial gene fragments.

### Box 2

### RABV-GLUE

RABV-GLUE is a 'sequence data resource' developed to support rabies elimination efforts by enabling efficient dissemination and utilisation of RABV sequence data using GLUE—a bioinformatics environment for managing and interpreting virus sequence data [15,19]. GLUE supports development of virus-specific 'projects' comprising curated sets of sequences, genome feature annotations, alignments, reference clades and phylogenies with associated metadata, with options to upload sequences to GenBank (Fig 2). Loading projects into the GLUE 'engine' creates a relational database representing complex semantic links between data items so computational analyses can be precisely and widely replicated.

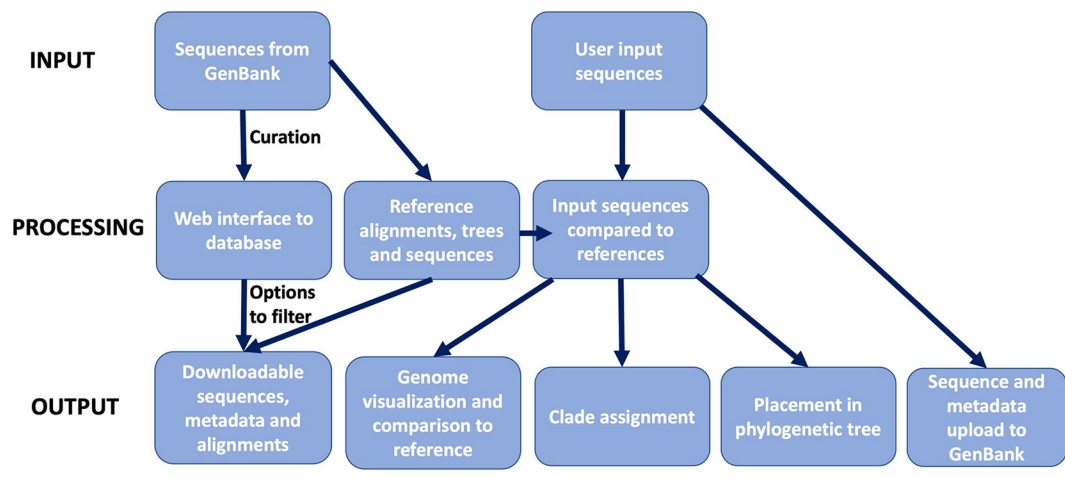

**Fig 2. Schematic of the online capabilities of RABV-GLUE.**

RABV-GLUE can be used to obtain sequences from GenBank and has options to filter and download sequences of interest, and for phylogenetic interpretation of user input sequences. The metadata for these sequences has been manually curated to correct mistakes and fill in information missing from GenBank to allow users access to cleaner, more

documented sequence data. As additional corrections are identified, RABV-GLUE will be curated live. RABV-GLUE projects can be installed locally on all commonly used computing platforms and are fully containerised via Docker [20]. GLUE's command layer provides a mechanism for retrieving and analysing data by coordinating interactions between the RABV-GLUE database and commonly used bioinformatics software tools (e.g. MAAFT, RAXML). This allows experienced bioinformaticians to quickly establish local RABV sequence databases and integrate these resources into existing bioinformatic pipelines, tailoring functionalities to their specific needs. Hosting the RABV-GLUE project in an openly accessible online version control system (e.g., GitHub) provides a mechanism for managing its ongoing development by multiple collaborators, following practices established in the software industry. GLUE projects can also be developed into interactive, user-friendly web services through a graphical user interface. An online interface to RABV-GLUE is available at http://rabv-glue.cvr.gla.ac.uk/ (Fig 2). Here we use sequence data and classifications from RABV-GLUE, to provide the basis for the development of an updated classification system which will be integrated into RABV-GLUE, enhancing its capacity to provide detailed, high resolution lineage information about user input sequences.

## Lineage designation

Sequences were aligned using MAFFT with the FFT-NS-2 algorithm [21]. Maximum Likelihood trees were constructed using IQTREE2 with model selection [22] and 100 bootstrap replicates [23]. Ancestral sequence reconstruction was then undertaken using Treetime ancestral [24].

A custom R function for lineage designation was developed which requires the tree, corresponding alignment, ancestral sequences and metadata. This returns summary statistics about each sequence, its lineage designation, and details about each of the designated lineages. Lineage defining nodes are identified according to thresholds on bootstrap support ($>$70) and cluster size ($>$10 descendants of $>$95% coverage at genome level or of the gene of interest, excluding gaps and ambiguous bases). For each node still in consideration, the ancestral sequence is extracted. To define a new lineage, there must be at least one common mutation between all descendants that is different to the ancestral sequence. Having at least 10 descendants also helps to ensure the robustness of designations, preventing the incorrect designation of lineages as a result of sequencing errors. The algorithm for lineage designation is summarised in Fig 3. The parameters for designation of partial genome sequences were refined to optimise the comparability with the whole genome designations, testing various bootstrap support thresholds and numbers of sequences needed to designate a lineage.

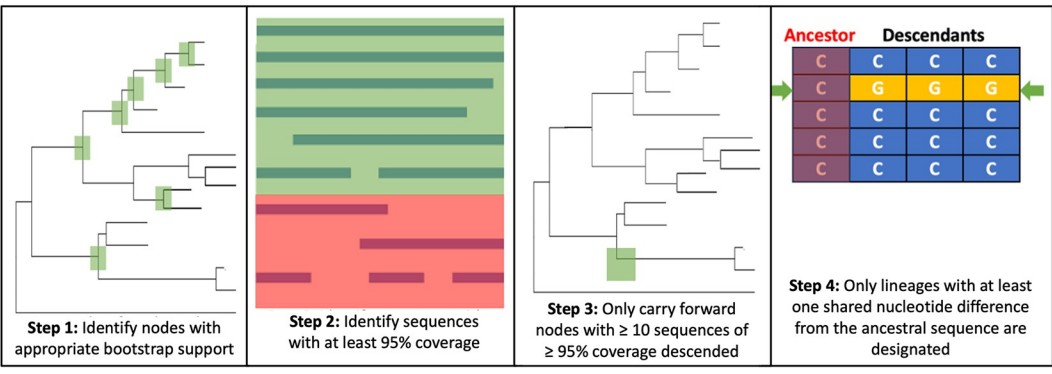

**Fig 3. Summary of lineage designation steps.**

Existing phylogenetic groupings are already in place for RABV [13,25] and sequences are automatically defined by these clades in RABV-GLUE which acts as a useful baseline for further classification. Subsequent MAD DOG lineage designations are named according to the rules in Rambaut et al. [18], starting from lineage A1 (which in this instance would be Cosmopolitan A1). Any lineages descended from this become A1.1 or A1.2 etc and any descended from these become A1.1.1, A1.2.1 etc. After 3 iterations a lineage becomes a new major lineage–A1.1.1.1 = B1.

## Lineage assignment

Once the designation step has been performed to construct a set of reference sequences with defined lineages, new sequences can be compared to the reference set and assigned to lineages. New sequences (imported individually or in bulk) are added to the existing reference alignment using MAFFT and for each new sequence the two closest references are identified and used to transfer a lineage assignment.

## Emerging and undersampled lineages

The requirement for a lineage to have 10 sequences was used to ensure that sequencing errors do not translate to artificial lineages. However, this criterion may result in emerging lineages, or undersampled lineages being initially overlooked. Therefore, we also include both an Emerging or Undersampled Lineages, and a Singletons of Interest output.

For emerging/undersampled lineages, any nodes that fulfil all the requirements for defining a lineage but only have between 5 and 9 sequences, not the 10 required for a full lineage designation, are extracted. To ensure that lineages are distinct, and not just a cluster of sequences within the parent lineage, the patristic distance between the defining node and its parent node are calculated. If this distance exceeds the 95th percentile of patristic distances between sequences within the parent lineage, the node is marked as defining a potentially emerging or undersampled lineage. Furthermore, to ensure the lineage is emerging/undersampled, the country and year of collection for each sequence in the potentially emerging lineage are extracted. If all the sequences in the potentially emerging lineage are from the same country within a 5-year period, the lineage is marked as emerging, named with an '_Ex' suffix (where x is a number indicating multiple distinct emerging/undersampled lineages within the same parent lineage), and added to the 'Emerging or Undersampled Lineages' table.

To identify singletons of interest, branch lengths are extracted for each tip and the longest 5% are identified. For each of the tips that have the longest 5% of branch lengths, their closest related sequence is identified. The patristic distance between the query sequence and its closest relative are calculated. If the distance exceeds the 95th percentile of patristic distances between sequences within the parent lineage, the sequence is marked as a singleton of interest. Information about the number and metadata of these sequences is collated for each lineage and added to the 'Singletons of Interest' table, to identify which may be of interest to investigate further.

## MAD DOG

MAD DOG has been developed as a command-line based tool to undertake all the steps for lineage designation and assignment outlined above, given a set of input sequences (and corresponding metadata for designation). The assignment tool also outputs information about the assigned lineage including which countries it has been sampled in and when the first and most recent sequences for that lineage were recorded. Additionally, this can be implemented through an R package (MADDOG), full details of which can be found at DOI: 10.5281/zenodo.5503916.

MAD DOG was used to designate lineages for sequence subsets from the Cosmopolitan clade, with its performance tested on whole and partial genome sequences. The resulting

designations were explored on a global scale and on a specific geographic area (Tanzania) to test functionality and interpret lineage assignments.

## Results

### Cosmopolitan dataset

The Cosmopolitan dataset comprised 650 WGS, excluding vaccine strains, spanning 46 countries from 1950–2018. The N gene data comprised an additional 1958 sequences spanning an additional 25 countries from 1939–2019. Running the lineage designation on Cosmopolitan WGS resulted in 52 lineages, more than a 2-fold increase in definition compared to the 22 previously defined sub-clades (Fig 4). The baseline MAD DOG designation agreed with the previously determined global phylogenetic groupings, with the exception of clusters containing <10 sequences (a required threshold for MAD DOG lineage designation). However, our tool provided a much deeper charac-terisation beyond the subclade level [13]. Some subclades are split into multiple lineages providing increased definition; four showed at least a 4-fold increase in definition, and one showed a 10-fold increase, in line with sampling density (Fig 4A and Table 1). The AF1b subclade (purple in Figs 4 and 5) is split into 10 lineages, and the ME1a (orange) subclade is split into 4, indicating more than a 4-fold increase in definition in both cases. Some subclades show no further definition due to the small number of sequences available to represent this subset of RABV diversity.

**Table 1. Details of numbers of lineages and sequences available for each Cosmopolitan subclade at whole genome and N gene level.**

| Subclade | Whole genome | | | N gene | | |
|---|---|---|---|---|---|---|
| | Lineages | Number of sequences | Year of first sequence | Lineages | Number of sequences | Year of first sequence |
| AF1a | 2 | 24 | 1984 | 10 | 377 | 1982 |
| AF1b | 10 | 251 | 1981 | 10 | 471 | 1981 |
| AF1c | 0 | 3 | 1986 | 0 | 6 | 1986 |
| AF4 | 0 | 8 | 1950 | 1 | 13 | 1950 |
| AM1 | 0 | 4 | 1982 | 6 | 58 | 1981 |
| AM2a | 1 | 10 | 1991 | 4 | 136 | 1990 |
| AM2b | 0 | 2 | 2009 | 0 | 24 | 1986 |
| AM3a | 0 | 5 | 1986 | 20 | 172 | 1985 |
| AM3b | 0 | 4 | 1986 | 10 | 108 | 1986 |
| AM4 | 0 | 4 | 1974 | 1 | 19 | 1974 |
| CA1 | 1 | 25 | 1974 | 9 | 224 | 1974 |
| CA2 | 1 | 15 | 1993 | 7 | 106 | 1993 |
| CA3 | 0 | 7 | 1991 | 6 | 45 | 1991 |
| CE | 1 | 15 | 1990 | 1 | 19 | 1985 |
| EE | 1 | 15 | 1986 | 1 | 47 | 1977 |
| ME1a | 4 | 101 | 1976 | 4 | 195 | 1976 |
| ME1b | 0 | 2 | 1993 | 0 | 7 | 1993 |
| ME2 | 4 | 38 | 1989 | 6 | 66 | 1989 |
| NEE | 6 | 71 | 1986 | 6 | 78 | 1985 |
| WE | 1 | 14 | 1986 | 1 | 148 | 1974 |
| YUGCOW | 0 | 2 | 1978 | 0 | 6 | 1978 |
| YUGFOX | 0 | 1 | 1972 | 0 | 12 | 1972 |

Lineage designation was performed on an N gene subset of the same data (the 650 sequences used for WGS designation) to allow a direct comparison between lineage designations at different

levels of sequence data resolution. This resulted in a 25% overall decrease compared to the definition achieved using WGS, highlighting that the sequence length impacts the designations. The reduced definition applies to some subclades more than others. Various numbers of minimum sequences and minimum bootstrap support values to define a lineage were tested on the N gene subset and the WGS designations to optimise the comparability of the two. Once the parameters for designation at N gene level were refined, a minimum of 10 sequences and bootstrap support of 70 was used for this subset, and applied to all available Cosmopolitan N gene sequences; resulting in an increase from 650 to 2608 sequences. Applying this modified algorithm to the N gene sequences resulted in lineage designations that did not entirely match up with the WGS designations, due to differences in sequence numbers and tree topology. This caused problems when attempting to analyse and compare lineages, and therefore some extra steps were required to ensure the N gene designations corresponded to the WGS designations. N gene sequences from any subclades that were not seen in the WGS designation, such as AM3a (Table 1), were extracted and lineage designation was run as normal on these. For those subclades that were split further into lineages at WGS level, the N gene sequences first underwent lineage assignment using the WGS designations as a reference. N gene lineages could then be designated, using the WGS lineage assignment as a starting point. This resulted in an additional 44 designations above the WGS designations due to the increased volume of sequences (Figs 4 and 5, Table 1), highlighting how data volume and coverage also affects the designations.

24 emerging lineages were identified, with multiple lineages emerging within some lineages, for example AM3a_A1 and CA1_A1 (S1 and S4 Tables). 16 of the designated lineages are identified as being potentially undersampled (S5 Table). Some of these contain multiple singleton sequences, such as 4 singletons within the AM3b_A1.1.1 lineage, indicating there may be multiple descendent lineages missed due to undersampling.

The geographic distribution of lineages can be seen in Fig 5, with full details in S1 Table. It appears that Europe, North America and Southern Africa are well represented. However, only the Cosmopolitan clade is included here and therefore the overall global RABV sequencing coverage of RABV lineages cannot be assessed from this alone.

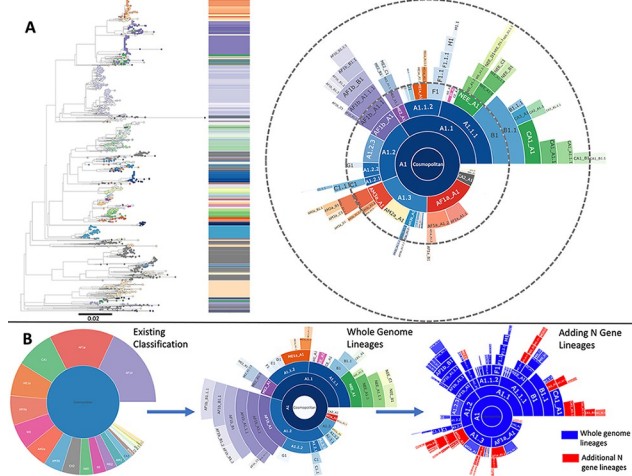

**Fig 4. Lineage designations of whole genome and N gene sequences from the RABV Cosmopolitan clade. A:** Maximum likelihood tree showing the lineage positions, rooted using an outgroup of 10 Asian SEA1b sequences, with scale indicating substitutions per site per year, and hierarchical relationships with dashed lines indicating 5 lineage iterations (n = 2608 total sequences for tree and sunburst plot, combing both WGS and N gene sequences). **B:** Progression of classification definition from 22 previously defined subclades **(left)** to 52 lineages using only whole genome sequences **(middle)** to 147 lineages adding N gene sequences, with lineages seen at whole genome level in blue and additional lineages only seen at N gene level in red **(right)**.

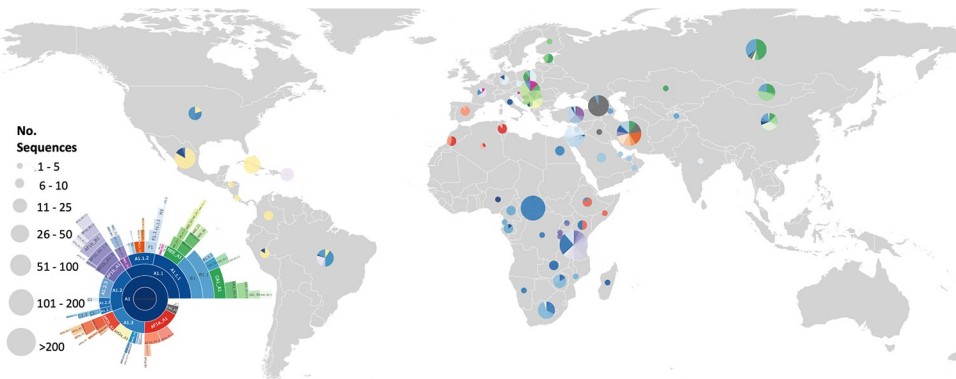

**Fig 5. Distribution of RABV lineages from the Cosmopolitan clade determined from WGS and N gene sequences.**
Circle size indicates the number of sequences (sum of both WGS and N gene sequences), with circles plotted to the centroid of each country. Sunburst plot from Fig 4A included to indicate colour scheme. Base layer of map from Natural Earth (https://www.naturalearthdata.com/).

WGS capture greater variation, and are able to differentiate between samples that are identical at a partial genome level [16]. In this study, although all sequences were unique at the whole genome level, only 80% of these were unique at an N gene level (n = 509 from 1958 N gene sequences). While whole genomes offer better phylogenetic resolution, the N gene is commonly targeted by diagnostic laboratories providing a greater number of sequences and wider spatio-temporal coverage to detect and define lineages not captured by the still limited number of WGS. In total, there are 1469 WGS of RABV (650 from the Cosmopolitan clade) from 82 countries available on RABV-GLUE but 7304 N gene sequences (1958 from the Cosmopolitan clade) from 109 countries. This highlights the volume of additional data available when using partial genome sequences. However, the proportion of studies sequencing RABV whole genomes is increasing, as are studies using at least N gene length sequences instead of smaller fragments (S1 Fig).

## Box 3

**Table 2. Case Studies using the MADDOG lineage classification system.**

| Example 1—Identifying the origin of cases | Example 2—Reconstructing historical RABV spread |
|---|---|
| The higher resolution achieved through this classification system can be used to 'zoom in' on unusual cases and look at the historical and geographical context of a lineage. Here, we illustrate the example of a human rabies case (GenBank accession: KC737850) reported from the USA in 2011 [26], where dog-associated rabies has been eliminated but wildlife variants are endemic.<br><br>The case report explains that this individual moved from Brazil to the USA, and many years later developed rabies [26]. In this situation the most likely cause of rabies was either exposure to wildlife rabies in the USA, or exposure to rabies in Brazil, which seemed unlikely given the long incubation period. Sequence data shows that the latter scenario is correct (Fig 6A), with the RABV sequence from the patient showing high similarity to dog RABV sequences in Brazil as reported by Boland et al. [26]. Using additional N gene sequences, we can examine this lineage in more detail. A sequence from Peru in 1999 (accession: KU938904) (Fig 6A), was listed in RABV-GLUE as being from the common vampire bat, *Desmodus rotundus* but the original GenBank record states 'livestock case infected by bat'. In this case it appears to have been misassigned as a bat host by GLUE's automatic curation from Genbank. However, the identification of a dog lineage in a bat is unusual. Follow-up with researchers from the original study revealed this was mislabelled on Genbank and was in fact most likely a livestock case infected by a dog [29]. These mislabelled entries are subsequently updated in RABV-GLUE. In this way, the metadata for sequences available in RABV-GLUE is constantly updated and corrected, providing additional and improved information beyond what is available on GenBank. The lineage classification system allows resolution of these unusual cases, and identification of sequences that are incorrectly labelled. | Rabies was first documented in Tanzania in the 1930's [27]. Although present in North Africa for centuries, the virus is thought to have become endemic and spread within sub-Saharan Africa following European colonisation in the twentieth century [27]. The literature suggests at least two historical introductions to Tanzania; from the North and the South of the country [28]. With the classification system, it is possible to identify spread over time on a finer scale, and therefore evaluate potential support for these theories.<br><br>The lineage A1.2.1 appears to support the spread of RABV into Tanzania from southern Africa (Fig 6B), with the first sequence from this ancestral lineage found in South Africa in 1981 (KX148103). By 1992 the lineage was detected in Zimbabwe (KT336434), as part of a study looking at potential incursions between South Africa and Zimbabwe [30], before being sequenced in Tanzania in 2010 [28]. This tracing of the lineage demonstrates the movement route across the continent has been followed at least once historically. Routine monitoring also allows us to identify that the lineage has persisted in South Africa, as the sequence MT454644 from 2017 is designated to the A1.2.1 lineage [31]. Additional information provided by inclusion of N gene data shows the lineage was also detected in Zambia from 1999, likely due to spread from neighbouring Zimbabwe. |

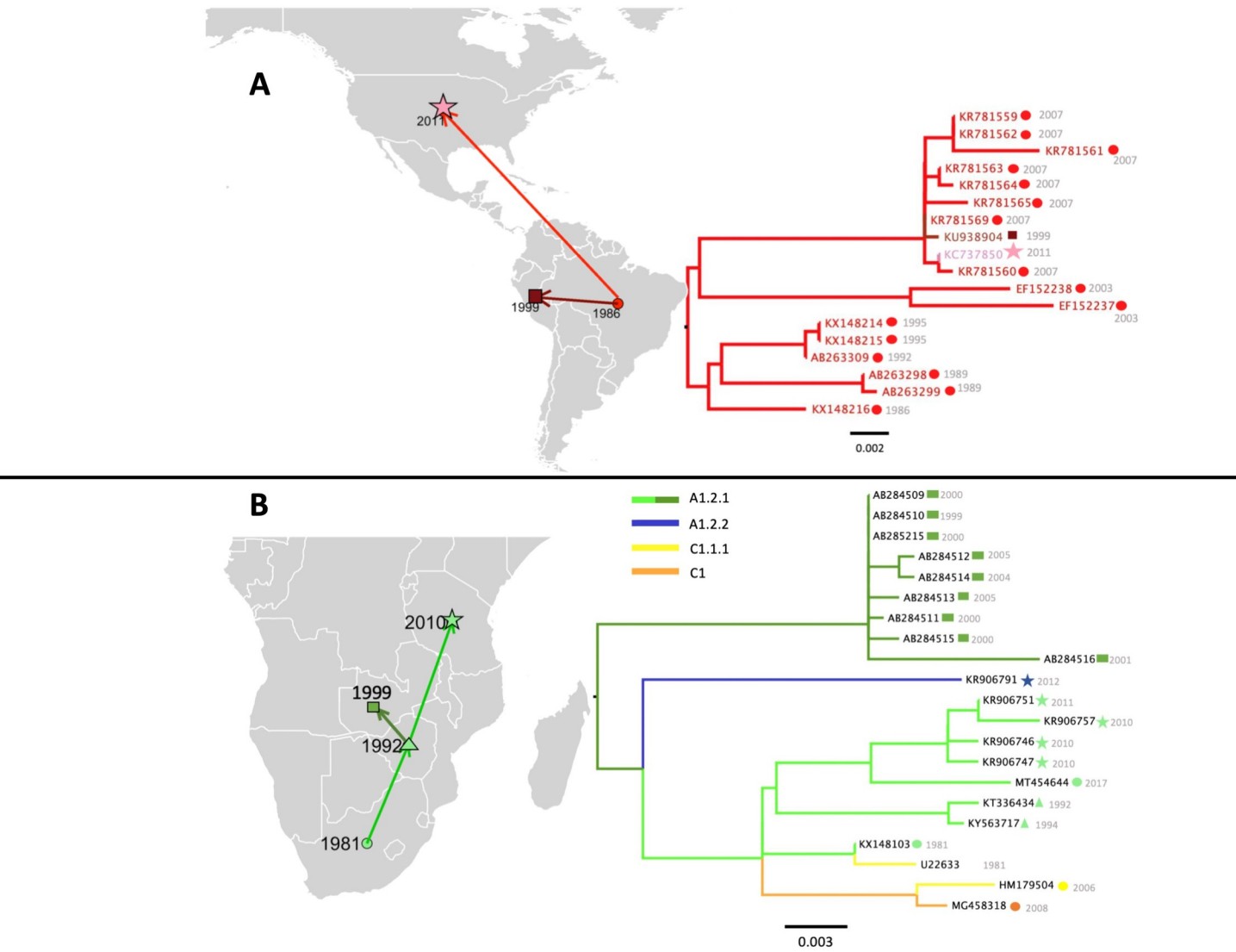

**Fig 6. Case studies identifying an imported human rabies case and reconstructing historical spread of RABV into Tanzania.** Dated points represent first records of the lineage in the country, with arrows suggesting the likely source of introductions. Scale indicates substitutions per site per year. Base layer of maps from Natural Earth (https://www.naturalearthdata.com/). **A: Left**: Countries where AM3a_A1.3 has been isolated over time. Bright red = information from WGS lineage designation, darker red = additional information from N gene lineage designation. The star indicates the human rabies case. **Right**: Maximum likelihood Subtree of AM3a_A1.3 lineage, with icons to indicate country of origin, **B: Left**: Countries where lineage A1.2.1 has been isolated over time. Bright green = information from WGS lineage designation, darker green = additional information from N gene lineage designation. The star indicates case of interest. **Right**: Maximum likelihood subtree of A1.2.1 lineage and descendents, with icons to indicate country of origin, corresponding to map, and dated tips.

## Local lineage designation case study—Tanzania

The Tanzania dataset comprised 224 WGS from 1996–2018. Running the lineage designation on these sequences alone, i.e. without wider geographic context, limited the designation of less commonly sampled lineages that may be more prevalent in other countries (S2 Table). The inclusion of sequences from Eastern and Southern Africa (known to belong to the same global subclade) resulted in an additional lineage designation that could not be defined using Tanzania sequences alone. Therefore, additional relevant context to provide an informative reference set for initial lineage designation proved to be important for

consistent, higher resolution lineage designations. This revealed 15 lineages present in Tanzania (S3 Table) from the Cosmopolitan clade, in contrast to only 14 defined with Tanzania data alone. Five lineages are specific to the Serengeti District and the remaining 10 are present in multiple areas (Fig 7). Of the 15 lineages, 13 have been seen in the Serengeti District. In 2010, there were 11 lineages circulating that had been detected in the Serengeti District. By 2017 following considerable dog vaccination efforts in the district but not in the wider region, only 6 were still evident.

There do not appear to be any singletons of interest present in Tanzania, however there is a lineage that may be undersampled; Cosmopolitan_A1.2.2_E1, for which 5 sequences are present from Tanzania between 2003–2004 (S4 and S5 Tables). Additional, more recent sequences are likely to show this lineage has become a fully designated lineage.

The increased use of whole genome sequencing is pronounced in Tanzania (S2 Fig); most Tanzanian sequences available on RABV-GLUE for the last decade are whole genome, with all sequences since 2005 being at least N gene.

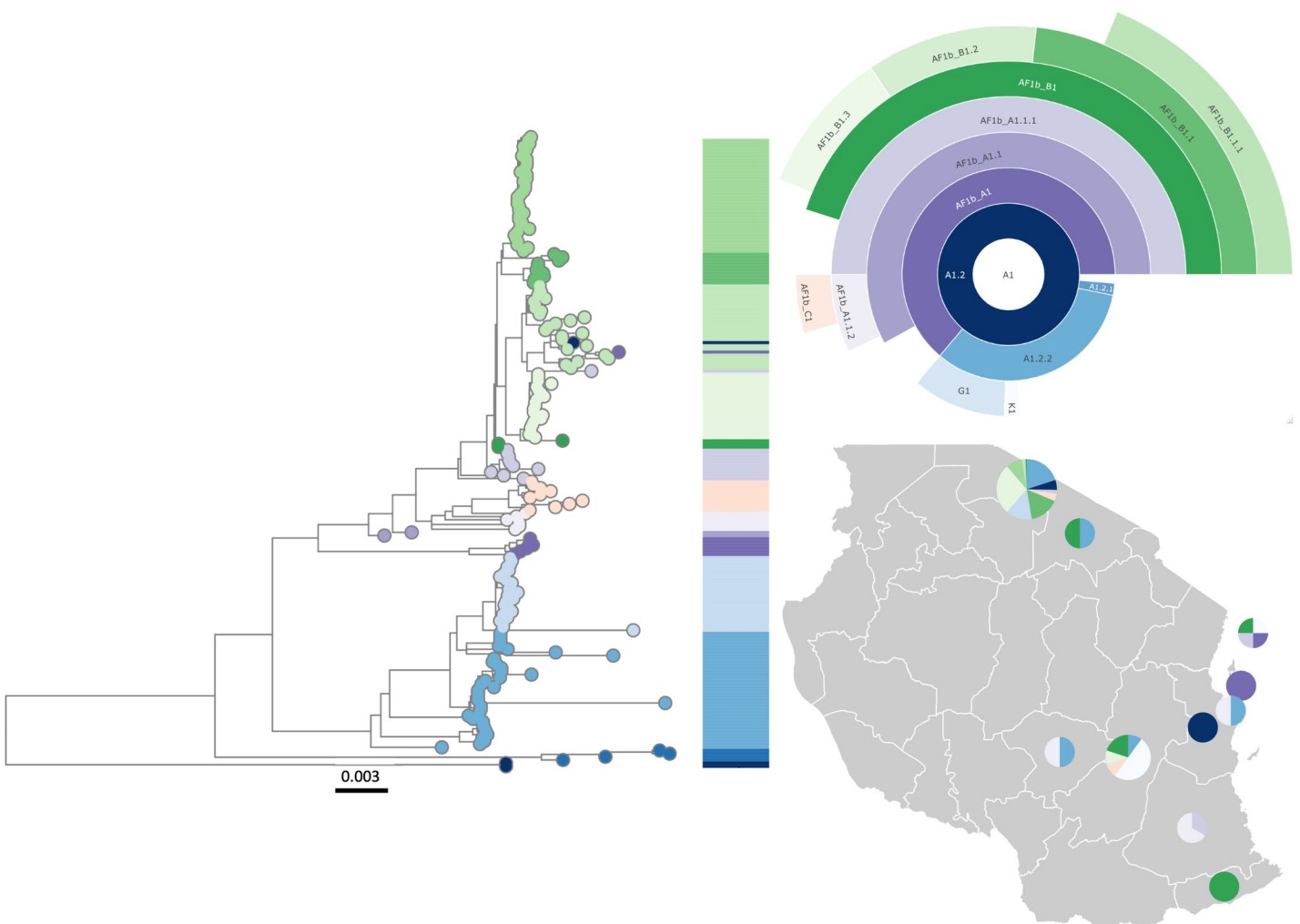

**Fig 7. Lineage designations of 224 rabies virus genomes from Tanzania.** Maximum likelihood tree with scale in substitutions per site per year, hierarchical relationships, and distribution of lineages. Sequences obtained from RABV-GLUE. Base layer of map from GADM (https://gadm.org/download_country.html).

## Discussion

We aimed to apply the dynamic nomenclature system proposed by Rambaut et al. (2020) to RABV and to investigate how this classification system could be useful in surveillance to support control and elimination programmes. Implementing this classification system allows for much greater resolution of circulating RABV lineages at both the whole genome and N gene levels. Whole genome level designations and assignments provide most resolution (doubling the resolution of analysis possible, with an over 10-fold increase in some areas) and accurate assignments. However, unlike SARS-CoV-2, which is a new pathogen with limited accumulated diversity over a recent time frame, rabies virus is an endemic pathogen that has circulated in some areas for decades. Therefore, our analysis highlighted the importance of considering partial genome data (N gene sequences) to incorporate the considerable historical data available for RABV (and more generally to other endemic pathogens with partial genome data). The case studies and examples presented here illustrate how higher resolution lineage designations enable greater epidemiological interpretation, for example, in identifying the origins of cases and for surveillance to monitor lineage introductions and extinctions as elimination is approached (Box 3, Table 2).

There are challenges in adapting the universal classification system developed for SARS-CoV-2 to RABV [18]. The spatial and temporal density of sequences is very different. More than 4 million WGS from over 185 countries are available for SARS-CoV-2 in a time span of less than two years [32,33], whereas RABV sequences span 7 decades but are much sparser. Moreover, SARS-CoV-2 lineages were designated from the emergence of this pathogen in human populations and have been updated as variation has accumulated. In contrast, the RABV lineage designation is retrospective; accounting for greater diversity and time. As the naming system calls for a new major lineage after 4 iterations of minor lineages (A1.1.1.1 = B1), the names do not fully reflect the depth of diversity and evolutionary time frame (e.g. that B1 is a descendent of A1), which can make it hard to see lineage turnover. For this reason, the full evolutionary path of each lineage is listed as an output i.e. for the AF1a_A1.2 lineage, AF1a_A1<-A1<-Cosmopolitan is listed (S1 Table). Likewise, it can be difficult to identify co-circulating lineages with this system as sequences are assigned to later iterations. For example, although A1 may no longer appear to be circulating, enough variation may have accumulated to designate more recent sequences to descendant lineage A1.1. However, these limitations are primarily to do with the naming of lineages, and analyses should be designed to incorporate the iterative nature of lineage designations, to represent the full extent of viral diversity. Additionally, the lineage designation steps implemented into the MAD DOG tool are not specific to RABV. The tool can therefore be used for any virus with just an input set of sequences and metadata. This may be valuable for other viruses with existing naming systems that need updating to show the appropriate definition for surveillance purposes, as the tool can incorporate existing naming systems and build from these. Additionally, we caution that the phylogenetic analyses reported here are not time-measured. MAD DOG is not intended as a tool for detailed phylodynamic analysis and more sophisticated inference, using BEAST or other analyses that incorporate the molecular clock are required to address important evolutionary and ecological questions e.g [12,34–36]. MADDOG is designed to serve as a rapid tool to provide a starting point for further investigations, assessing the number and diversity of sequenced lineages. Thus MADDOG may provide valuable insight into the direction subsequent analyses should focus on or rapidly actionable information in the event of identifying sources of incursions (or sustained but undetected local transmission) and for informing control policies accordingly.

One of the most recent and widely cited global phylogenomic analyses of RABV (Troupin et al. (2016)), used 321 WGS to capture and represent a level of geographic and temporal

RABV diversity that was previously unexplored, covering all 6 major clades (Cosmopolitan, Asian, Africa-2, Africa-3, Arctic and Indian Subcontinent) [13]. However, since then publicly available RABV genomes have tripled and the way in which genomic surveillance is utilised has advanced from (usually) retrospective studies defining broad phylogenetic patterns, towards informing local control efforts in real-time. Therefore, we used this study as the baseline for designating coarse phylogenetic groups (using the existing clade names) but showcase the added value of our high resolution lineage designation tool for more specific epidemiological investigations. Sequence data has become increasingly accessible and we use 650 WGS just from the Cosmopolitan clade (over 50% of total dog-related RABV WGS available), to provide an improved characterisation of RABV diversity that supports our aim of interpreting sequence data for local and national surveillance. This approach follows from other local studies, for example, Brunker et al. (2015) identify 5 lineages circulating in Tanzania using phylogenetic methods and genetic distances, showing their co-circulation and identifying historical introductions and human-mediated movement of infected animals [28]. However, different studies define and name lineages in different ways. A study from Cameroon used geographic area as part of its definition of viral lineages [37] while Talbi et al. [38] identify 8 "subtypes" within the Africa 2 clade (named A-H) defined by phylogenetic placement with Bayesian posterior support; thus splitting the subclade further, but not to a high resolution [38]. These examples highlight how the lack of a universal lineage definition and naming system makes comparison between studies challenging, as the scale and method for lineage assignment and designation are not comparable.

An area that highlights the use of MAD DOG is the whole genome analysis of the Africa-1b subclade (AF1b), which is seen to be present in 9 African countries, with sequences spanning 1981–2018. Eight of the 10 AF1b lineages have only been seen in Tanzania, while the remainder have been detected in multiple countries (S1 Table). This subclade has good definition due to the large number of sequences, with over 10% of these WGS from a large South African study [31] and several studies in Tanzania [10,28,39] that generated over 80% of the available AF1b WGS. When incorporating N gene data for AF1b, the limited diversity compared to WGS means additional N gene sequences do not translate into designated lineages, as also evident in the ME1a and WE subclades (Fig 4 and Table 1). The definition of designated lineages therefore depends both on the breadth of the data (the length of the sequences) and the depth of the data (number of sequences). Overall, a 25% decrease in definition, and in some instances up to 50%, is seen when using N gene sequences alone compared to WGS of the same sequence set. Despite the lower resolution provided by N gene data, the large volume of partial genome sequences makes their inclusion essential for representing previously identified diversity. As with the WGS designated lineages, those subclades not seen are due to having fewer than 10 sequences available (Table 1).

A significant area of difference between the whole genome and N gene designations, that highlights the importance of using all available data, is found in the Africa-1a (AF1a) subclade. When only using whole genomes (21 WGS), the AF1a subclade splits into 2 lineages seen across multiple countries. With all the available N gene data (310 sequences) AF1a is split into an additional 3 lineages, with distinct lineages present in Ethiopia (1) and Tunisia (1). The first N gene sequence in the AF1a subclade is from 1982, whereas the first WGS is from 2 years later. Although the difference here is only two years, in other subclades, such as AM2b, the earliest available N gene sequences are from two decades earlier, thus excluding N gene sequences would exclude considerable historical data, and limit inference about historical patterns of dispersal.

Of the 24 emerging/undersampled lineages, some appear to be nascent lineages that did not persist, such as Cosmopolitan_B1.1_E2 which contains 7 sequences from France in 1991, but

no further sequences. It may be that this lineage was emerging but halted due to an outbreak that was controlled. Alternatively, it may be that this is an undersampled lineage that was only not designated as a full lineage due to lack of sequences. In contrast, Cosmopolitan CA1_B1.1_E1 contains 7 sequences from Mongolia detected in 2017–2018 and therefore is likely to be an emerging lineage that could still be identified as a full lineage with the addition of more recent sequence data (S4 Table). Of the 16 singletons of interest, singletons are seen both in deeper lineages, such as AF1b_A1, and more terminal lineages, such as AM3b_A1.1.1. These singletons could be indicative of sequencing errors, or could be true significant diversity, perhaps indicating very sparse sequencing failing to capture the extent of the diversity within the lineage or the very beginning of a new, divergent lineage. Lineage AM3b_A1.1.1 contains 4 singletons of interest (all sequenced from Brazil in 2006), which could indicate multiple distinct descendent lineages that cannot be designated due to undersampling (S5 Table). These singletons likely represent sparse sequencing of samples during this time. Investigating undersampled contemporary lineages in detail is beyond the scope of this study, but could provide insight into where viral diversity is accumulating or overlooked, and monitoring may allow the identification of areas where new lineages are emerging.

Applying the MAD DOG lineage designation on a local scale allows in-country transmission routes and persistence of lineages to be seen at new depth. Tanzania is used as an example, which has a large number of WGS with detailed metadata. This provides greater inference to identify the uses and potential issues of this classification system on a local scale, such as local lineage dynamics and identification of areas needing improved vaccination efforts. 152 of the 205 Tanzanian sequences are from the Serengeti District, with 14 of the 15 Tanzanian lineages being detected there. This reflects greater sampling density enabling detection of circulating lineages, which also may be present elsewhere but remain undetected due to limited sampling. The sequence data point to the impact of improved dog vaccination in Serengeti district [40], with the apparent reduction in lineages from 11 in 2010 to just 6 by 2017. However, more sequences from 2017 onwards will be needed to determine whether lineage extinctions have truly occurred, or if limited ongoing sequencing has affected the detection of older lineages that are persisting in the district or elsewhere in the country.

While we provide proof of concept by applying MAD DOG to only the Cosmopolitan clade, the lineage classification and nomenclature system that we have developed can be easily extended to update all the RABV clades, and has potential to be extended to other viruses. The development of RABV-GLUE, and the incorporation of the updated classifications, provides an accessible platform for interpreting RABV sequence data with detailed genotyping that incorporates these high resolution lineage designations even for users unfamiliar with genomic analysis. This functionality is important in rapid sequencing, and to inform policy, as it should allow detailed, accessible interpretations about a sequence to be given quickly, which may for example be used to identify sustained transmission or the source of an incursion. As well as the improved access to and interpretation of sequence data through RABV-GLUE, the increased resolution provided by this updated system can be used to better understand the spread and persistence of RABV and resulting lineage dynamics, as highlighted by the examples presented here. These tools should prove valuable for monitoring the progress of rabies control programmes as they strive to achieve the 2030 goal.

## Supporting information

**S1 Table. Lineage Information for WGS and N gene lineages.** Details of the 147 designated lineages from 2608 rabies virus whole genome (10,000nt) and N gene (1300nt) sequences from the Cosmopolitan clade, obtained from RABV-GLUE. Includes all countries sequences

assigned to each lineage have been seen in, the first and most recent collection years of those sequences, the number of sequences assigned to each lineage for both whole genome and N gene sequences. The evolutionary path of each lineage is also listed.
(CSV)

**S2 Table. Tanzania Lineage Designation.** Details of the 14 designated lineages from lineage designation run exclusively on whole genome (10000nt) rabies virus sequences from Tanzania, obtained from RABV-GLUE. Includes all places sequences assigned each lineage have been seen in, the first and most recent collection years of those sequences, the maximum and mean patristic distance of all sequences in each lineage and the lineages descended from it, and the number of sequences assigned to each lineage. Where place = NA, no information is publicly available for sequence locations. Mara region contains the Serengeti District.
(DOCX)

**S3 Table. Tanzania Subset of Global Lineage Information.** Details of the 15 global MAD DOG lineages that have been detected in Tanzania. Includes all places sequences assigned to each lineage have been seen in, the first and most recent collection years of those sequences and the number of sequences assigned to each lineage.
(DOCX)

**S4 Table. Cosmopolitan Emerging or Undersampled Lineages.** Details of potentially emerging or undersampled lineages in the Cosmopolitan clade that do not yet have enough sequences to be defined as a new lineage. Lineages are named with an '_Ex' suffix (where x is a number indicating multiple distinct emerging/undersampled lineages within the same parent lineage). The number of tips for each lineage, as well as the country and time period the sequences are from are also listed, as is the patristic distance between the node defining the lineage and its parent node.
(DOCX)

**S5 Table. Cosmopolitan Singletons of Interest.** Details of singleton sequences with long branch lengths (longest 5%), indicating potentially undersampled lineages, sequencing errors, or newly emerging divergent lineages that need to be monitored. The number of singleton sequences in each lineage is listed, plus collection years and counties of those singleton sequences where this information is publicly available.
(DOCX)

**S1 Fig. Global Sequence Length Maps and Time Series. Left:** Map of collection locations of all available whole genome and N gene sequences. Base layer of map from Natural Earth (https://www.naturalearthdata.com/). **Right:** Time series of number of whole genome, N gene, and shorter sequences collected.
(TIF)

**S2 Fig. Tanzanian Sequence Length Time Series.** Timeseries of the number of whole genome, N gene, and shorter partial genome sequences from Tanzania available on RABV-GLUE.
(TIF)

## Acknowledgments

We would like to thank Roman Biek and Daniel Streicker for their constructive feedback, Afrida Mukaddas for help and support in developing RABV-GLUE.

## Author Contributions

**Conceptualization:** Kathryn Campbell.

**Data curation:** Kathryn Campbell.

**Formal analysis:** Kathryn Campbell.

**Investigation:** Kathryn Campbell.

**Methodology:** Kathryn Campbell, Verity Hill, Aine O'Toole, Andrew Rambaut.

**Software:** Kathryn Campbell, Robert J. Gifford, Joshua Singer.

**Supervision:** Katie Hampson, Kirstyn Brunker.

**Visualization:** Kathryn Campbell.

**Writing – original draft:** Kathryn Campbell.

**Writing – review & editing:** Kathryn Campbell, Robert J. Gifford, Katie Hampson, Kirstyn Brunker.

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
