## [Decision Letter · Decision Letter 0]

6 Jan 2022

Dear Miss Campbell,

Thank you very much for submitting your manuscript "Making Genomic Surveillance Deliver: A Lineage Classification and Nomenclature System to Inform Rabies Elimination" for consideration at PLOS Pathogens. As with all papers reviewed by the journal, your manuscript was reviewed by members of the editorial board and by several independent reviewers. In light of the reviews (below this email), we would like to invite the resubmission of a significantly-revised version that takes into account the reviewers' comments.

Thank you for this submission, and again my apologies for the extreme delays noted in my prior correspondence.

All three reviewers were enthusiastic about this work. All three make suggestions for improving the presentation of the data and demonstrating the utility of your new system. Reviewer 1 and 3 have important constructive critiques that would need to be addressed in a revision.

We cannot make any decision about publication until we have seen the revised manuscript and your response to the reviewers' comments. Your revised manuscript is also likely to be sent to reviewers for further evaluation.

Sincerely,

Adam Lauring

Section Editor

PLOS Pathogens

Kasturi Haldar

Editor-in-Chief

PLOS Pathogens

orcid.org/0000-0001-5065-158X

Michael Malim

Editor-in-Chief

PLOS Pathogens

orcid.org/0000-0002-7699-2064

Thank you for this submission, and again my apologies for the extreme delays noted in my prior correspondence.

All three reviewers were enthusiastic about this work. All three make suggestions for improving the presentation of the data and demonstrating the utility of your new system. Reviewer 1 and 3 have important constructive critiques that would need to be addressed in a revision.

Reviewer's Responses to Questions

**Part I - Summary**

Reviewer #1: The manuscript „Making Genomic Surveillance Deliver: A Lineage Classification and Nomenclature System to Inform Rabies Elimination“ by Campbell and coauthors describes a high resolution phylogenetic classification system for rabies to identify incursions and transmission routes. The proposed model is based on the experiences with SARS-CoV-2 and could also be used for other epizootics/enzootics.

Optimization of the classification of outbreak strains is important and improvements seem helpful.

A major strength of the presented workflow is the inclusion of WGS and partial sequences (like the N gene) as well as a more structured and more detailed classification which could help to compare genetic epidemiology analysis of different working groups in the future.

However, it remains unclear, what the advantages for the genomic epidemiology and control measures are. Statements about the need of improved classifications should be more clear and more detailed. Today, analysis of transmission routes is done for rabies in numerous studies (e.g. Denise A Marston, Daniel L Horton, Javier Nunez, Richard J Ellis, Richard J Orton, Nicholas Johnson, Ashley C Banyard, Lorraine M McElhinney, Conrad M Freuling, Müge Fırat, Nil Ünal, Thomas Müller, Xavier de Lamballerie, Anthony R Fooks, Genetic analysis of a rabies virus host shift event reveals within-host viral dynamics in a new host, Virus Evolution, Volume 3, Issue 2, July 2017, vex038, https://doi.org/10.1093/ve/vex038). In those studies, phylogenetic or BEAST analyses allowed highly detailed data and conclusions.

In the manuscript, it must therefore be made even clearer what the advantages of the new methodology are. Overall, the structure of the manuscript takes some getting used to and should be reconsidered.

Reviewer #2: Campbell et al. develop a system to classify rabies virus sequences into lineages, adapting the existing PANGO lineage method developed for SARS-CoV-2. The authors describe this new method and provide example cases that highlight the utility of more fine-grained classifications for rabies surveillance and transmission analysis. Overall, the paper is clearly written, the classification system makes sense, and the examples provided are a nice addition that really highlights the method concretely. Additionally, the comparison between resolution from WGS compared to N gene is helpful. I have only minor comments.

Reviewer #3: In this work, Campbell et al. applied a dynamic lineage classification and nomenclature system previously used for SARS-CoV-2 on rabies virus (RABV) sequence data, specifically for the Cosmopolitan clade of viruses. Compared to the present system which only provides a high-level subspecies classification, the finer delineations produced by the authors facilitate epidemiological interpretations such as origins identification and surveillance of global as well as local circulating strains. The manuscript is generally well-written with a clear presentation on how the newly applied system could be used to partially describe the spatiotemporal dynamics of RABV.

**Part II – Major Issues: Key Experiments Required for Acceptance**

Reviewer #1: 1. Please explain the term „…we find an over 200% increase in the definition…“. What does the 200% refer to (- Line 57 and 318)? Is this changing the resolution of the analysis possible today or does this just change and unify the classifications?

2. How was it proven, that there were areas of persistence and transmission that were not previously apparent? Please provide data and explanations (- Line 58)

3. Would a „standard analysis“ using Blast, MEGA and BEAST result in the same outcome (as e.g. in Figure 6)? Please provide a direct comparison to convince the reader to use the new classification tool. What is the advantage of the strain designation for this type of analysis of a standard tree and BEAST analysis (e.g. Figure 7)? Please provide more detailed explanations/data of the advantages.

4. It is not clear why the proposed new classification system is supporting the „Zero by 30“ aim. Is this view supported by WHO/FAO and rabies reference laboratories?

Reviewer #2: (No Response)

Reviewer #3: However, there are a couple of concerns I hope the authors will find helpful in improving their work:

1) I am presuming that the phylogenetic trees shown in Figure 4 are patristic distance trees and not temporally-resolved ones. While there isn’t a scale to denote the branch lengths, a fair proportion of tips appear as singletons with extended branch lengths from the common ancestor of the virus clade they have been assigned to. These tips likely represent undersampled diversity. While these singletons may share common mutations with the larger clades that they are assigned to, they should also encode multiple mutations that would otherwise be deemed genetically distinct should there be more similar sequences sampled. Such singletons are less likely to appear for SARS-CoV-2 given its recent emergence and the unprecedented volume of sequence data that have been generated in substantially shorter timescales. As such, I feel that there is a need to tweak the current lineage classification system to flag and distinguish these singletons, which in itself may help inform on where there might be surveillance gaps.

Although the authors did not address this specifically, they note that the sparse availability of RABV sequences meant that the new nomenclature “do not fully reflect the depth of diversity and evolutionary time frame” of the viruses. The authors then made the argument that this limitation would only affect the naming of lineages, and that “analyses should be designed to incorporate the iterative nature of lineage designations”. However, if the aim of this proposed system is to be a new standard to capture underlying evolutionary and/or epidemiological processes which the authors assert, more care should be taken into the precision of what evolutionary/epidemiological bounds are embedded within these names.

2) I am also curious as to the choice of outgroup used to root the tree. Is the tree currently rooted by temporal structure? If so, how robust is the temporal structure? If the authors claim that the new nomenclature system can help identify case origins and reconstruct historical spread, the rooting choice isn’t a trivial one as it will impact interpretations on the directionality of evolutionary events and reconstruction of ancestral states.

**Part III – Minor Issues: Editorial and Data Presentation Modifications**

Reviewer #1: 1. Box 1: what is the difference/relation of clade and lineage should be explained. Please define also clade designation and clade assignment.

2. Figure 3/step 3: how to deal with new outbreaks with novel lineages and less than 10 sequences in the early phase?

3. The keywords listed should be checked. Words like phylogenetics or lineage are missing.

4. The boxes seem to be missplaced in the introduction section.

5. Figure 3 seems to be more part of the results section

6. Figure 3/step 4: Single nucleotide changes are used for classification. How to exclude sequencing errors as a reason for new „lineages“? How robust is the analysis concerning sequencing errors?

7. Define „cosmopolitan clade“

8. Several of the supplemental datasets (figures/tabes) are not mentioned within the text.

Reviewer #2: 1. The authors mention in the Methods that MADDOG is available via an R package, which is great. However, the authors do not provide a link to the code, installation instructions, an example dataset, or any sort of documentation. Because this paper primarily introduces a new method, it would be helpful if the authors provided a direct link to the method, even if it is just a github page. I was not able to find such a page from a google search, and I think it would be helpful to potential users to provide at minimum a link to a web page with some documentation, an example dataset, and installation instructions.

2. Figure 5 is a bit challenging to interpret as currently colored and described in the legend. From reading the legend, I had inferred that when a given country had 2 circles overlaid on its centroid, that the darker/more vibrant one represented lineages inferred from WGS, while the lighter one represented those inferred from N gene sequences. However, some countries have multiple overlaid circles that are all light (Brazil, US, Mexico). For these countries, it was not clear to me what the interior vs. exterior circles represented. Additionally, the colors in the less vibrant circles were challenging to distinguish. I would suggest clarifying this in the legend, and perhaps altering the figure in some way to make the colors easier to distinguish for the light circles.

3. The case examples detailed in figures 6 and 7 were excellent and very helpful. However, showing some additional data would be helpful in showing how the sequence data and lineage designations allow for the inferences shown in the map and arrow graphics. For figure 6, it would be helpful to include the tree or at least a partial snippet of the tree, so that the data used for the inference shown on the map is clear. For figure 7b, it would be helpful to show on country-level information for the tips shown on the tree. This could be added as additional text in the strain name, as shapes or colors for the tips, or some other method. Having this information directly on the tree would make it more clear how the tree was used to infer the transmission history shown in 7a.

This is just a compliment: I thought it was very cool that the authors followed up with the original authors of the case report described in example 1 to confirm their inferences! That was a fun example to read.

Reviewer #3: Lastly, a minor point - can the authors provide a main or supplementary figure showing the difference between the current and proposed lineage designation? I think it will be impactful to show how the new nomenclature system yield more informative delineations compared to the currently used one.

PLOS authors have the option to publish the peer review history of their article (what does this mean?). If published, this will include your full peer review and any attached files.

Reviewer #1: No

Reviewer #2: No

Reviewer #3: No
---

## [Decision Letter · Decision Letter 1]

30 Mar 2022

Dear Miss Campbell,

We are pleased to inform you that your manuscript 'Making Genomic Surveillance Deliver: A Lineage Classification and Nomenclature System to Inform Rabies Elimination' has been provisionally accepted for publication in PLOS Pathogens.

Best regards,

Adam S. Lauring

Section Editor

PLOS Pathogens

Adam Lauring

Section Editor

PLOS Pathogens

Kasturi Haldar

Editor-in-Chief

PLOS Pathogens

orcid.org/0000-0001-5065-158X

Michael Malim

Editor-in-Chief

PLOS Pathogens

orcid.org/0000-0002-7699-2064

Reviewer Comments (if any, and for reference):

Reviewer's Responses to Questions

**Part I - Summary**

Reviewer #1: The revised version has much improved and all critical/open points were well addressed.

Reviewer #2: The authors have been quite responsive to reviewer comments. I had only minor comments to begin with, and they have all been appropriately addressed.

Reviewer #3: The authors have responded to all of my previous comments satisfactorily. It is good to see the integration between the WGS and N genes by using the latter for further lineage delineation based on comments from other reviewers and great to see that the authors took care to differentiate sampled viruses that may be emerging/under-sampled/likely resulted from sequencing errors. Nice piece of work, congratulations.

**Part II – Major Issues: Key Experiments Required for Acceptance**

Reviewer #1: No further modifications are requested from my site.

Reviewer #2: (No Response)

Reviewer #3: I have no other major issues.

**Part III – Minor Issues: Editorial and Data Presentation Modifications**

Reviewer #1: No further minor issues.

Reviewer #2: (No Response)

Reviewer #3: None.

PLOS authors have the option to publish the peer review history of their article (what does this mean?). If published, this will include your full peer review and any attached files.

Reviewer #1: No

Reviewer #2: No

Reviewer #3: No

---

## [Editor Report · Acceptance letter]

25 Apr 2022

Dear Miss Campbell,

We are delighted to inform you that your manuscript, "Making Genomic Surveillance Deliver: A Lineage Classification and Nomenclature System to Inform Rabies Elimination," has been formally accepted for publication in PLOS Pathogens.

Best regards,

Kasturi Haldar

Editor-in-Chief

PLOS Pathogens

orcid.org/0000-0001-5065-158X

Michael Malim

Editor-in-Chief

PLOS Pathogens

orcid.org/0000-0002-7699-2064